# STRUCTURED SEQUENCE MODELING WITH GRAPH CONVOLUTIONAL RECURRENT NETWORKS

**Youngjoo Seo**
EPFL, Switzerland
`youngjoo.seo@epfl.ch`

**Michaël Defferrard**
EPFL, Switzerland
`michael.defferrard@epfl.ch`

**Pierre Vandergheynst**
EPFL, Switzerland
`pierre.vandergheynst@epfl.ch`

**Xavier Bresson**
EPFL, Switzerland
`xavier.bresson@epfl.ch`

## ABSTRACT

This paper introduces Graph Convolutional Recurrent Network (GCRN), a deep learning model able to predict structured sequences of data. Precisely, GCRN is a generalization of classical recurrent neural networks (RNN) to data structured by an arbitrary graph. Such structured sequences can represent series of frames in videos, spatio-temporal measurements on a network of sensors, or random walks on a vocabulary graph for natural language modeling. The proposed model combines convolutional neural networks (CNN) on graphs to identify spatial structures and RNN to find dynamic patterns. We study two possible architectures of GCRN, and apply the models to two practical problems: predicting moving MNIST data, and modeling natural language with the Penn Treebank dataset. Experiments show that exploiting simultaneously graph spatial and dynamic information about data can improve both precision and learning speed.

## 1  INTRODUCTION

Many real-world data can be cast as structured sequences, with spatio-temporal sequences being a special case. A well-studied example of spatio-temporal data are videos, where succeeding frames share temporal and spatial structures. Many works, such as Donahue et al. (2015); Karpathy & Fei-Fei (2015); Vinyals et al. (2015), leveraged a combination of CNN and RNN to exploit such spatial and temporal regularities. Their models are able to process possibly time-varying visual inputs for variable-length prediction. These neural network architectures consist of combining a CNN for visual feature extraction followed by a RNN for sequence learning. Such architectures have been successfully used for video activity recognition, image captioning and video description.

More recently, interest has grown in properly fusing the CNN and RNN models for spatio-temporal sequence modeling. Inspired by language modeling, Ranzato et al. (2014) proposed a model to represent complex deformations and motion patterns by discovering both spatial and temporal correlations. They showed that prediction of the next video frame and interpolation of intermediate frames can be achieved by building a RNN-based language model on the visual words obtained by quantizing the image patches. Their highest-performing model, recursive CNN (rCNN), uses convolutions for both inputs and states. Shi et al. (2015) then proposed the convolutional LSTM network (convLSTM), a recurrent model for spatio-temporal sequence modeling which uses 2D-grid convolution to leverage the spatial correlations in input data. They successfully applied their model to the prediction of the evolution of radar echo maps for precipitation nowcasting.

The spatial structure of many important problems may however not be as simple as regular grids. For instance, the data measured from meteorological stations lie on a irregular grid, i.e. a network of heterogeneous spatial distribution of stations. More challenging, the spatial structure of data may not even be spatial, as it is the case for social or biological networks. Eventually, the interpretation that sentences can be regarded as random walks on vocabulary graphs, a view popularized by Mikolov et al. (2013), allows us to cast language analysis problems as graph-structured sequence models.

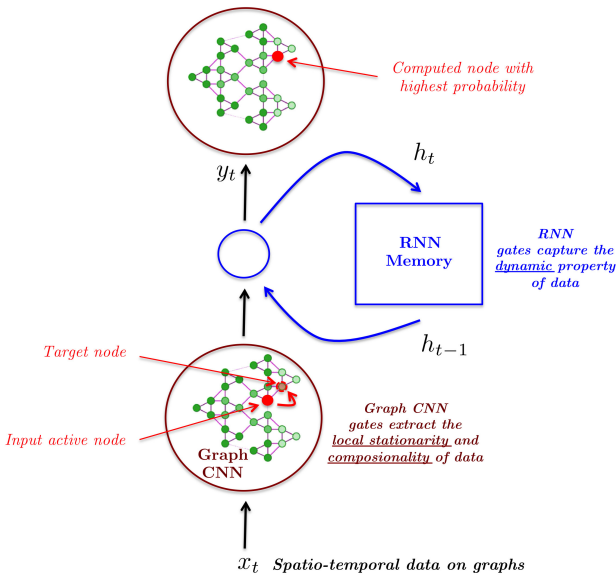
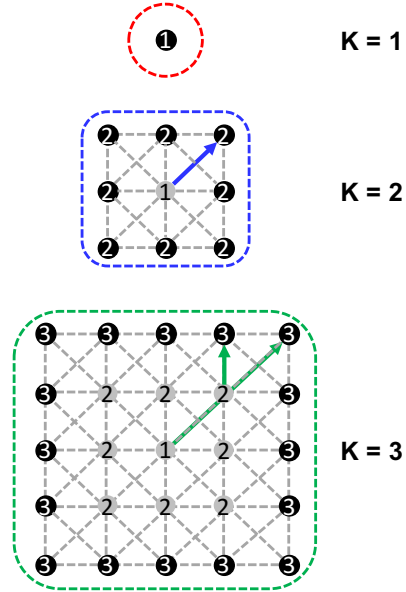

Figure 1: Illustration of the proposed GCRN model for spatio-temporal prediction of graph-structured data. The technique combines at the same time CNN on graphs and RNN. RNN can be easily exchanged with LSTM or GRU networks.

Figure 2: Illustration of the neighborhood on an 8-nearest-neighbor grid graph. Isotropic spectral filters of support $K$ have access to nodes at most at $K-1$ hops.

This work leverages on the recent models of Defferrard et al. (2016); Ranzato et al. (2014); Shi et al. (2015) to design the GCRN model for modeling and predicting time-varying graph-based data. The core idea is to merge CNN for graph-structured data and RNN to identify simultaneously meaningful spatial structures and dynamic patterns. A generic illustration of the proposed GCRN architecture is given by Figure 1.

## 2 PRELIMINARIES

### 2.1 STRUCTURED SEQUENCE MODELING

Sequence modeling is the problem of predicting the most likely future length-$K$ sequence given the previous $J$ observations:

$$\hat{x}_{t+1}, \ldots, \hat{x}_{t+K} = \underset{x_{t+1}, \ldots, x_{t+K}}{\arg\max} \ P(x_{t+1}, \ldots, x_{t+K} | x_{t-J+1}, \ldots, x_t), \tag{1}$$

where $x_t \in \mathbf{D}$ is an observation at time $t$ and $\mathbf{D}$ denotes the domain of the observed features. The archetypal application being the $n$-gram language model (with $n = J + 1$), where $P(x_{t+1} | x_{t-J+1}, \ldots, x_t)$ models the probability of word $x_{t+1}$ to appear conditioned on the past $J$ words in the sentence (Graves, 2013).

In this paper, we are interested in special structured sequences, i.e. sequences where features of the observations $x_t$ are not independent but linked by pairwise relationships. Such relationships are universally modeled by weighted graphs.

Data $x_t$ can be viewed as a graph signal, i.e. a signal defined on an undirected and weighted graph $\mathcal{G} = (\mathcal{V}, \mathcal{E}, A)$, where $\mathcal{V}$ is a finite set of $|\mathcal{V}| = n$ vertices, $\mathcal{E}$ is a set of edges and $A \in \mathbb{R}^{n \times n}$ is a weighted adjacency matrix encoding the connection weight between two vertices. A signal $x_t : \mathcal{V} \to \mathbb{R}^{d_x}$ defined on the nodes of the graph may be regarded as a matrix $x_t \in \mathbb{R}^{n \times d_x}$ whose column $i$ is the $d_x$-dimensional value of $x_t$ at the $i^{th}$ node. While the number of free variables in a structured sequence of length $K$ is in principle $\mathcal{O}(n^K d_x{}^K)$, we seek to exploit the structure of the space of possible predictions to reduce the dimensionality and hence make those problems more tractable.

## 2.2 Long Short-Term Memory

A special class of recurrent neural networks (RNN) that prevents the gradient from vanishing too quickly is the popular long short-term memory (LSTM) introduced by Hochreiter & Schmidhuber (1997). This architecture has proven stable and powerful for modeling long-range dependencies in various general-purpose sequence modeling tasks (Graves, 2013; Srivastava et al., 2015; Sutskever et al., 2014). A fully-connected LSTM (FC-LSTM) may be seen as a multivariate version of LSTM where the input $x_t \in \mathbb{R}^{d_x}$, cell output $h_t \in [-1, 1]^{d_h}$ and states $c_t \in \mathbb{R}^{d_h}$ are all vectors. In this paper, we follow the FC-LSTM formulation of Graves (2013), that is:

$$
\begin{aligned}
i &= \sigma(W_{xi}x_t + W_{hi}h_{t-1} + w_{ci} \odot c_{t-1} + b_i), \\
f &= \sigma(W_{xf}x_t + W_{hf}h_{t-1} + w_{cf} \odot c_{t-1} + b_f), \\
c_t &= f_t \odot c_{t-1} + i_t \odot \tanh(W_{xc}x_t + W_{hc}h_{t-1} + b_c), \\
o &= \sigma(W_{xo}x_t + W_{ho}h_{t-1} + w_{co} \odot c_t + b_o), \\
h_t &= o \odot \tanh(c_t),
\end{aligned}
\tag{2}
$$

where $\odot$ denotes the Hadamard product, $\sigma(\cdot)$ the sigmoid function $\sigma(x) = 1/(1 + e^{-x})$ and $i, f, o \in [0, 1]^{d_h}$ are the input, forget and output gates. The weights $W_{x\cdot} \in \mathbb{R}^{d_h \times d_x}$, $W_{h\cdot} \in \mathbb{R}^{d_h \times d_h}$, $w_{c\cdot} \in \mathbb{R}^{d_h}$ and biases $b_i, b_f, b_c, b_o \in \mathbb{R}^{d_h}$ are the model parameters.[1] Such a model is called fully-connected because the dense matrices $W_{x\cdot}$ and $W_{h\cdot}$ linearly combine all the components of $x$ and $h$. The optional peephole connections $w_{c\cdot} \odot c_t$, introduced by Gers & Schmidhuber (2000), have been found to improve performance on certain tasks.

## 2.3 Convolutional Neural Networks on Graphs

Generalizing convolutional neural networks (CNNs) to arbitrary graphs is a recent area of interest. Two approaches have been explored in the literature: (i) a generalization of the spatial definition of a convolution (Masci et al., 2015; Niepert et al., 2016) and (ii), a multiplication in the graph Fourier domain by the way of the convolution theorem (Bruna et al., 2014; Defferrard et al., 2016). Masci et al. (2015) introduced a spatial generalization of CNNs to 3D meshes. The authors used geodesic polar coordinates to define convolution operations on mesh patches, and formulated a deep learning architecture which allows comparison across different meshes. Hence, this method is tailored to manifolds and is not directly generalizable to arbitrary graphs. Niepert et al. (2016) proposed a spatial approach which may be decomposed in three steps: (i) select a node, (ii) construct its neighborhood and (iii) normalize the selected sub-graph, i.e. order the neighboring nodes. The extracted patches are then fed into a conventional 1D Euclidean CNN. As graphs generally do not possess a natural ordering (temporal, spatial or otherwise), a labeling procedure should be used to impose it. Bruna et al. (2014) were the first to introduce the spectral framework described below in the context of graph CNNs. The major drawback of this method is its $\mathcal{O}(n^2)$ complexity, which was overcome with the technique of Defferrard et al. (2016), which offers a linear complexity $\mathcal{O}(|\mathcal{E}|)$ and provides strictly localized filters. Kipf & Welling (2016) took a first-order approximation of the spectral filters proposed by Defferrard et al. (2016) and successfully used it for semi-supervised classification of nodes. While we focus on the framework introduced by Defferrard et al. (2016), the proposed model is agnostic to the choice of the graph convolution operator $*_\mathcal{G}$.

As it is difficult to express a meaningful translation operator in the vertex domain (Bruna et al., 2014; Niepert et al., 2016), Defferrard et al. (2016) chose a spectral formulation for the convolution operator on graph $*_\mathcal{G}$. By this definition, a graph signal $x \in \mathbb{R}^{n \times d_x}$ is filtered by a non-parametric kernel $g_\theta(\Lambda) = \mathrm{diag}(\theta)$, where $\theta \in \mathbb{R}^n$ is a vector of Fourier coefficients, as

$$
y = g_\theta *_\mathcal{G} x = g_\theta(L)x = g_\theta(U\Lambda U^T)x = U g_\theta(\Lambda) U^T x \in \mathbb{R}^{n \times d_x},
\tag{3}
$$

where $U \in \mathbb{R}^{n \times n}$ is the matrix of eigenvectors and $\Lambda \in \mathbb{R}^{n \times n}$ the diagonal matrix of eigenvalues of the normalized graph Laplacian $L = I_n - D^{-1/2}AD^{-1/2} = U\Lambda U^T \in \mathbb{R}^{n \times n}$, where $I_n$ is the identity matrix and $D \in \mathbb{R}^{n \times n}$ is the diagonal degree matrix with $D_{ii} = \sum_j A_{ij}$ (Chung, 1997). Note that the signal $x$ is filtered by $g_\theta$ with an element-wise multiplication of its graph Fourier transform $U^T x$ with $g_\theta$ (Shuman et al., 2013). Evaluating (3) is however expensive, as the multiplication with $U$ is $\mathcal{O}(n^2)$. Furthermore, computing the eigendecomposition of $L$ might

---

[1] A practical trick is to initialize the biases $b_i$, $b_f$ and $b_o$ to one such that the gates are initially open.

be prohibitively expensive for large graphs. To circumvent this problem, Defferrard et al. (2016) parametrizes $g_\theta$ as a truncated expansion, up to order $K-1$, of Chebyshev polynomials $T_k$ such that

$$g_\theta(\Lambda) = \sum_{k=0}^{K-1} \theta_k T_k(\tilde{\Lambda}), \qquad (4)$$

where the parameter $\theta \in \mathbb{R}^K$ is a vector of Chebyshev coefficients and $T_k(\tilde{\Lambda}) \in \mathbb{R}^{n \times n}$ is the Chebyshev polynomial of order $k$ evaluated at $\tilde{\Lambda} = 2\Lambda/\lambda_{max} - I_n$. The graph filtering operation can then be written as

$$y = g_\theta *_\mathcal{G} x = g_\theta(L)x = \sum_{k=0}^{K-1} \theta_k T_k(\tilde{L})x, \qquad (5)$$

where $T_k(\tilde{L}) \in \mathbb{R}^{n \times n}$ is the Chebyshev polynomial of order $k$ evaluated at the scaled Laplacian $\tilde{L} = 2L/\lambda_{max} - I_n$. Using the stable recurrence relation $T_k(x) = 2xT_{k-1}(x) - T_{k-2}(x)$ with $T_0 = 1$ and $T_1 = x$, one can evaluate (5) in $\mathcal{O}(K|\mathcal{E}|)$ operations, i.e. linearly with the number of edges. Note that as the filtering operation (5) is an order $K$ polynomial of the Laplacian, it is $K$-localized and depends only on nodes that are at maximum $K$ hops away from the central node, the $K$-neighborhood. The reader is referred to Defferrard et al. (2016) for details and an in-depth discussion.

## 3    RELATED WORKS

Shi et al. (2015) introduced a model for regular grid-structured sequences, which can be seen as a special case of the proposed model where the graph is an image grid where the nodes are well ordered. Their model is essentially the classical FC-LSTM (2) where the multiplications by dense matrices $W$ have been replaced by convolutions with kernels $W$:

$$
\begin{aligned}
i &= \sigma(W_{xi} * x_t + W_{hi} * h_{t-1} + w_{ci} \odot c_{t-1} + b_i), \\
f &= \sigma(W_{xf} * x_t + W_{hf} * h_{t-1} + w_{cf} \odot c_{t-1} + b_f), \\
c_t &= f_t \odot c_{t-1} + i_t \odot \tanh(W_{xc} * x_t + W_{hc} * h_{t-1} + b_c), \\
o &= \sigma(W_{xo} * x_t + W_{ho} * h_{t-1} + w_{co} \odot c_t + b_o), \\
h_t &= o \odot \tanh(c_t),
\end{aligned}
\qquad (6)
$$

where $*$ denotes the 2D convolution by a set of kernels. In their setting, the input tensor $x_t \in \mathbb{R}^{n_r \times n_c \times d_x}$ is the observation of $d_x$ measurements at time $t$ of a dynamical system over a spatial region represented by a grid of $n_r$ rows and $n_c$ columns. The model holds spatially distributed hidden and cell states of size $d_h$ given by the tensors $c_t, h_t \in \mathbb{R}^{n_r \times n_c \times d_h}$. The size $m$ of the convolutional kernels $W_{h\cdot} \in \mathbb{R}^{m \times m \times d_h \times d_h}$ and $W_{x\cdot} \in \mathbb{R}^{m \times m \times d_h \times d_x}$ determines the number of parameters, which is independent of the grid size $n_r \times n_c$. Earlier, Ranzato et al. (2014) proposed a similar RNN variation which uses convolutional layers instead of fully connected layers. The hidden state at time $t$ is given by

$$h_t = \tanh(\sigma(W_{x2} * \sigma(W_{x1} * x_t)) + \sigma(W_h * h_{t-1})), \qquad (7)$$

where the convolutional kernels $W_h \in \mathbb{R}^{d_h \times d_h}$ are restricted to filters of size 1x1 (effectively a fully connected layer shared across all spatial locations).

Observing that natural language exhibits syntactic properties that naturally combine words into phrases, Tai et al. (2015) proposed a model for tree-structured topologies, where each LSTM has access to the states of its children. They obtained state-of-the-art results on semantic relatedness and sentiment classification. Liang et al. (2016) followed up and proposed a variant on graphs. Their sophisticated network architecture obtained state-of-the-art results for semantic object parsing on four datasets. In those models, the states are gathered from the neighborhood by way of a weighted sum with trainable weight matrices. Those weights are however not shared across the graph, which would otherwise have required some ordering of the nodes, alike any other spatial definition of graph convolution. Moreover, their formulations are limited to the one-neighborhood of the current node, with equal weight given to each neighbor.

Motivated by spatio-temporal problems like modeling human motion and object interactions, Jain et al. (2016) developed a method to cast a spatio-temporal graph as a rich RNN mixture which

essentially associates a RNN to each node and edge. Again, the communication is limited to directly connected nodes and edges.

The closest model to our work is probably the one proposed by Li et al. (2015), which showed stat-of-the-art performance on a problem from program verification. Whereas they use the iterative procedure of the Graph Neural Networks (GNNs) model introduced by Scarselli et al. (2009) to propagate node representations until convergence, we instead use the graph CNN introduced by Defferrard et al. (2016) to diffuse information across the nodes. While their motivations are quite different, those models are related by the fact that a spectral filter defined as a polynomial of order $K$ can be implemented as a $K$-layer GNN.[2]

## 4  PROPOSED GCRN MODELS

We propose two GCRN architectures that are quite natural, and investigate their performances in real-world applications in Section 5.

**Model 1.** The most straightforward definition is to stack a graph CNN, defined as (5), for feature extraction and an LSTM, defined as (2), for sequence learning:

$$
\begin{aligned}
x_t^{\text{CNN}} &= \text{CNN}_{\mathcal{G}}(x_t) \\
i &= \sigma(W_{xi} x_t^{\text{CNN}} + W_{hi} h_{t-1} + w_{ci} \odot c_{t-1} + b_i), \\
f &= \sigma(W_{xf} x_t^{\text{CNN}} + W_{hf} h_{t-1} + w_{cf} \odot c_{t-1} + b_f), \\
c_t &= f_t \odot c_{t-1} + i_t \odot \tanh(W_{xc} x_t^{\text{CNN}} + W_{hc} h_{t-1} + b_c), \\
o &= \sigma(W_{xo} x_t^{\text{CNN}} + W_{ho} h_{t-1} + w_{co} \odot c_t + b_o), \\
h_t &= o \odot \tanh(c_t).
\end{aligned}
\tag{8}
$$

In that setting, the input matrix $x_t \in \mathbb{R}^{n \times d_x}$ may represent the observation of $d_x$ measurements at time $t$ of a dynamical system over a network whose organization is given by a graph $\mathcal{G}$. $x_t^{\text{CNN}}$ is the output of the graph CNN gate. For a proof of concept, we simply choose here $x_t^{\text{CNN}} = W^{\text{CNN}} *_{\mathcal{G}} x_t$, where $W^{\text{CNN}} \in \mathbb{R}^{K \times d_x \times d_x}$ are the Chebyshev coefficients for the graph convolutional kernels of support $K$. The model also holds spatially distributed hidden and cell states of size $d_h$ given by the matrices $c_t, h_t \in \mathbb{R}^{n \times d_h}$. Peepholes are controlled by $w_{c\cdot} \in \mathbb{R}^{n \times d_h}$. The weights $W_{h\cdot} \in \mathbb{R}^{d_h \times d_h}$ and $W_{x\cdot} \in \mathbb{R}^{d_h \times d_x}$ are the parameters of the fully connected layers. An architecture such as (8) may be enough to capture the data distribution by exploiting local stationarity and compositionality properties as well as the dynamic properties.

**Model 2.** To generalize the convLSTM model (6) to graphs we replace the Euclidean 2D convolution $*$ by the graph convolution $*_{\mathcal{G}}$:

$$
\begin{aligned}
i &= \sigma(W_{xi} *_{\mathcal{G}} x_t + W_{hi} *_{\mathcal{G}} h_{t-1} + w_{ci} \odot c_{t-1} + b_i), \\
f &= \sigma(W_{xf} *_{\mathcal{G}} x_t + W_{hf} *_{\mathcal{G}} h_{t-1} + w_{cf} \odot c_{t-1} + b_f), \\
c_t &= f_t \odot c_{t-1} + i_t \odot \tanh(W_{xc} *_{\mathcal{G}} x_t + W_{hc} *_{\mathcal{G}} h_{t-1} + b_c), \\
o &= \sigma(W_{xo} *_{\mathcal{G}} x_t + W_{ho} *_{\mathcal{G}} h_{t-1} + w_{co} \odot c_t + b_o), \\
h_t &= o \odot \tanh(c_t).
\end{aligned}
\tag{9}
$$

In that setting, the support $K$ of the graph convolutional kernels defined by the Chebyshev coefficients $W_{h\cdot} \in \mathbb{R}^{K \times d_h \times d_h}$ and $W_{x\cdot} \in \mathbb{R}^{K \times d_h \times d_x}$ determines the number of parameters, which is independent of the number of nodes $n$. To keep the notation simple, we write $W_{xi} *_{\mathcal{G}} x_t$ to mean a graph convolution of $x_t$ with $d_h d_x$ filters which are functions of the graph Laplacian $L$ parametrized by $K$ Chebyshev coefficients, as noted in (4) and (5). In a distributed computing setting, $K$ controls the communication overhead, i.e. the number of nodes any given node $i$ should exchange with in order to compute its local states.

The proposed blend of RNNs and graph CNNs is not limited to LSTMs and is straightforward to apply to any kind of recursive networks. For example, a vanilla RNN $h_t = \tanh(W_x x_t + W_h h_{t-1})$

---

[2]The basic idea is to set the transition function as a diffusion and the output function such as to realize the polynomial recurrence, then stack $K$ of those. See Defferrard et al. (2016) for details.

| Architecture | Structure | Filter size | Parameters | Runtime | Test(w/o Rot) | Test(Rot) |
|---|---|---|---|---|---|---|
| FC-LSTM | N/A | N/A | 142,667,776 | N/A | 4832 | - |
| LSTM+CNN | N/A | $5 \times 5$ | 13,524,496 | 2.10 | 3851 | 4339 |
| LSTM+CNN | N/A | $9 \times 9$ | 43,802,128 | 6.10 | 3903 | 4208 |
| LSTM+GCNN | $knn = 8$ | $K = 3$ | 1,629,712 | 0.82 | 3866 | 4367 |
| LSTM+GCNN | $knn = 8$ | $K = 5$ | 2,711,056 | 1.24 | 3495 | 3932 |
| LSTM+GCNN | $knn = 8$ | $K = 7$ | 3,792,400 | 1.61 | **3400** | **3803** |
| LSTM+GCNN | $knn = 8$ | $K = 9$ | 4,873,744 | 2.15 | 3395 | 3814 |
| LSTM+GCNN | $knn = 4$ | $K = 7$ | 3,792,400 | 1.61 | 3446 | 3844 |
| LSTM+GCNN | $knn = 16$ | $K = 7$ | 3,792,400 | 1.61 | 3578 | 3963 |

Table 1: Comparison between models. Runtime is the time spent per each mini-batch in seconds. Test cross-entropies correspond to moving MNIST, and rotating and moving MNIST. LSTM+GCNN is Model 2 defined in (9). Cross-entropy of FC-LSTM is taken from Shi et al. (2015).

would be modified as

$$h_t = \tanh(W_x *_\mathcal{G} x_t + W_h *_\mathcal{G} h_{t-1}), \tag{10}$$

and a Gated Recurrent Unit (GRU) (Cho et al., 2014) as

$$
\begin{aligned}
z &= \sigma(W_{xz} *_\mathcal{G} x_t + W_{hz} *_\mathcal{G} h_{t-1}), \\
r &= \sigma(W_{xr} *_\mathcal{G} x_t + W_{hr} *_\mathcal{G} h_{t-1}), \\
\tilde{h} &= \tanh(W_{xh} *_\mathcal{G} x_t + W_{hh} *_\mathcal{G} (r \odot h_{t-1})), \\
h_t &= z \odot h_{t-1} + (1 - z) \odot \tilde{h}.
\end{aligned}
\tag{11}
$$

As demonstrated by Shi et al. (2015), structure-aware LSTM cells can be stacked and used as sequence-to-sequence models using an architecture composed of an encoder, which processes the input sequence, and a decoder, which generates an output sequence. A standard practice for machine translation using RNNs (Cho et al., 2014; Sutskever et al., 2014).

## 5 EXPERIMENTS

### 5.1 SPATIO-TEMPORAL SEQUENCE MODELING ON MOVING-MNIST

For this synthetic experiment, we use the moving-MNIST dataset generated by Shi et al. (2015). All sequences are 20 frames long (10 frames as input and 10 frames for prediction) and contain two handwritten digits bouncing inside a $64 \times 64$ patch. Following their experimental setup, all models are trained by minimizing the binary cross-entropy loss using back-propagation through time (BPTT) and RMSProp with a learning rate of $10^{-3}$ and a decay rate of 0.9. We choose the best model with early-stopping on validation set. All implementations are based on their Theano code and dataset.[3] The adjacency matrix $A$ is constructed as a k-nearest-neighbor (knn) graph with Euclidean distance and Gaussian kernel between pixel locations. For a fair comparison with Shi et al. (2015) defined in (6), all GCRN experiments are conducted with Model 2 defined in (9), which is the same architecture with the 2D convolution $*$ replaced by a graph convolution $*_\mathcal{G}$. To further explore the impact of the isotropic property of our filters, we generated a variant of the moving MNIST dataset where digits are also rotating (see Figure 4).

Table 1 shows the performance of various models: (i) the baseline FC-LSTM from Shi et al. (2015), (ii) the 1-layer LSTM+CNN from Shi et al. (2015) with different filter sizes, and (iii) the proposed LSTM+graph CNN(GCNN) defined in (9) with different supports $K$. These results show the ability of the proposed method to capture spatio-temporal structures. Perhaps surprisingly, GCNNs can offer better performance than regular CNNs, even when the domain is a 2D grid and the data is images, the problem CNNs were initially developed for. The explanation is to be found in the differences between 2D filters and spectral graph filters. While a spectral filter of support $K = 3$ corresponds to the reach of a patch of size $5 \times 5$ (see Figure 2), the difference resides in the isotropic nature of the former and the number of parameters: $K = 3$ for the former and $5^2 = 25$ for the later.

---

[3]http://www.wanghao.in/code/SPARNN-release.zip

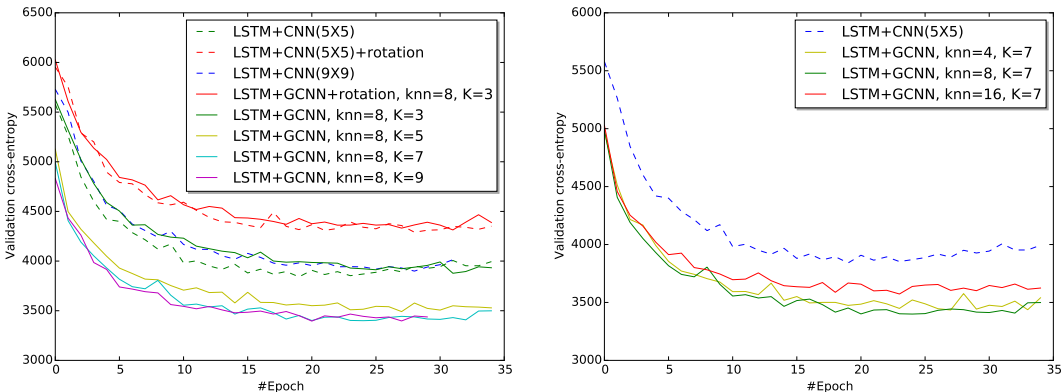

Figure 3: Cross-entropy on validation set: Left: performance of graph CNN with various filter support $K$. Right: performance w.r.t. graph construction.

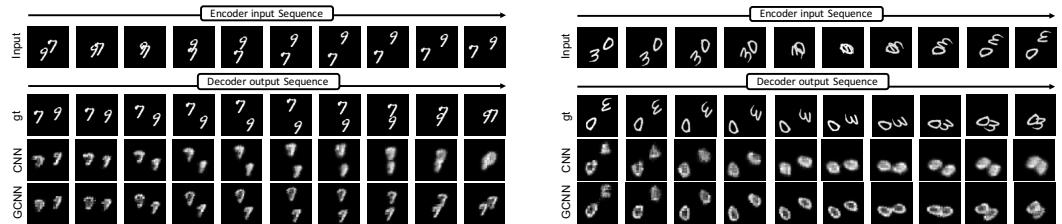

Figure 4: Qualitative results for moving MNIST, and rotating and moving MNIST. First row is the input sequence, second the ground truth, and third and fourth are the predictions of the LSTM+CNN($5 \times 5$) and LSTM+GCNN($knn = 8, K = 7$).

Table 1 indeed shows that LSTM+CNN($5 \times 5$) rivals LSTM+GCNN with $K = 3$. However, when increasing the filter size to $9 \times 9$ or $K = 5$, the GCNN variant clearly outperforms the CNN variant. This experiment demonstrates that graph spectral filters can obtain superior performance on regular domains with much less parameters thanks to their isotropic nature, a controversial property. Indeed, as the nodes are not ordered, there is no notion of an edge going up, down, on the right or on the left. All edges are treated equally, inducing some sort of rotation invariance. Additionally, Table 1 shows that the computational complexity of each model is linear with the filter size, and Figure 3 shows the learning dynamic of some of the models.

## 5.2 Natural Language Modeling on Penn Treebank

The Penn Treebank dataset has 1,036,580 words. It was pre-processed in Zaremba et al. (2014) and split[4] into a training set of 929k words, a validation set of 73k words, and a test set of 82k words. The size of the vocabulary of this corpus is 10,000. We use the gensim library[5] to compute a word2vec model (Mikolov et al., 2013) for embedding the words of the dictionary in a 200-dimensional space. Then we build the adjacency matrix of the word embedding using a 4-nearest neighbor graph with cosine distance. Figure 6 presents the computed adjacency matrix, and its 3D visualization. We used the hyperparameters of the small configuration given by the code[6] based on Zaremba et al. (2014): the size of the data mini-batch is 20, the number of temporal steps to unroll is 20, the dimension of the hidden state is 200. The global learning rate is 1.0 and the norm of the gradient is bounded by 5. The learning decay function is selected to be $0.5^{\max(0, \#epoch-4)}$. All experiments have 13 epochs, and dropout value is 0.75. For Zaremba et al. (2014), the input representation $x_t$ can be either the 200-dim embedding vector of the word, or the 10,000-dim one-hot representation of the word. For

---

[4]https://github.com/wojzaremba/lstm
[5]https://radimrehurek.com/gensim/models/word2vec.html

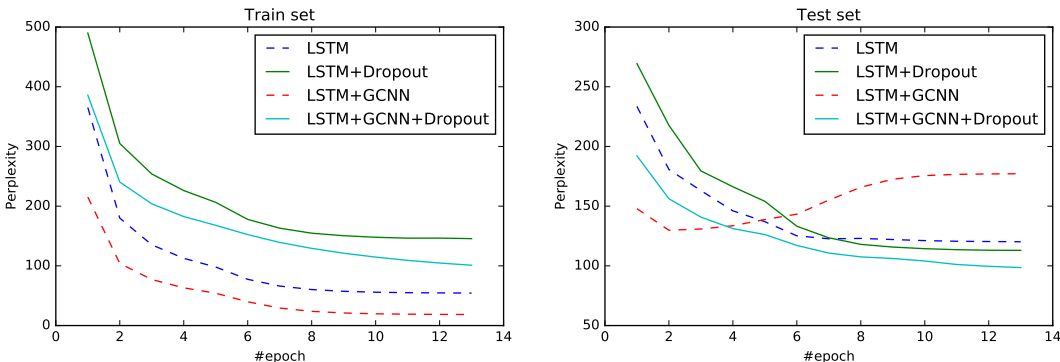

Figure 5: Learning dynamic of LSTM with and without graph structure and dropout regularization.

| Architecture | Representation | Parameters | Train Perplexity | Test Perplexity |
|---|---|---|---|---|
| Zaremba et al. (2014) code[6] | embedding | 681,800 | 36.96 | 117.29 |
| Zaremba et al. (2014) code[6] | one-hot | 34,011,600 | 53.89 | 118.82 |
| LSTM | embedding | 681,800 | 48.38 | 120.90 |
| LSTM | one-hot | 34,011,600 | 54.41 | 120.16 |
| LSTM, dropout | one-hot | 34,011,600 | 145.59 | 112.98 |
| GCRN-M1 | one-hot | 42,011,602 | 18.49 | 177.14 |
| GCRN-M1, dropout | one-hot | 42,011,602 | 114.29 | **98.67** |

Table 2: Comparison of models in terms of perplexity. Zaremba et al. (2014) code[6] is ran as benchmark algorithm. The original Zaremba et al. (2014) code used as input representation for $x_t$ the 200-dim embedding representation of words, computed here by the gensim library[5]. As our model runs on the 10,000-dim one-hot representation of words, we also ran Zaremba et al. (2014) code on this representation. We re-implemented Zaremba et al. (2014) code with the same architecture and hyperparameters. We remind that GCRN-M1 refers to GCRN Model 1 defined in (8).

our models, the input representation is a one-hot representation of the word. This choice allows us to use the graph structure of the words.

Table 2 reports the final train and test perplexity values for each investigated model and Figure 5 plots the perplexity value vs. the number of epochs for the train and test sets with and without dropout regularization. Numerical experiments show:

1. Given the same experimental conditions in terms of architecture and *no* dropout regularization, the standalone model of LSTM is more accurate than LSTM using the spatial graph information (120.16 vs. 177.14), extracted by graph CNN with the GCRN architecture of Model 1, Eq. (8).

2. However, using dropout regularization, the graph LSTM model overcomes the standalone LSTM with perplexity values 98.67 vs. 112.98.

3. The use of spatial graph information found by graph CNN speeds up the learning process, and overfits the training dataset in the absence of dropout regularization. The graph structure likely acts a constraint on the learning system that is forced to move in the space of language topics.

4. We performed the same experiments with LSTM and Model 2 defined in (9). Model 1 significantly outperformed Model 2, and Model 2 did worse than standalone LSTM. This bad performance may be the result of the large increase of dimensionality in Model 2, as the dimension of the hidden and cell states changes from 200 to 10,000, the size of the vocabulary. A solution would be to downsize the data dimensionality, as done in Shi et al. (2015) in the case of image data.

---

[6]`https://github.com/tensorflow/tensorflow/blob/master/tensorflow/
models/rnn/ptb/ptb_word_lm.py`

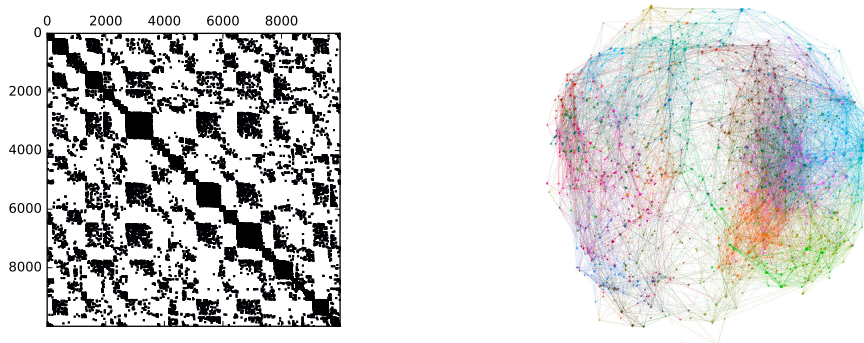

Figure 6: Left: adjacency matrix of word embeddings. Right: 3D visualization of words' structure.

## 6    CONCLUSION AND FUTURE WORK

This work aims at learning spatio-temporal structures from graph-structured and time-varying data. In this context, the main challenge is to identify the best possible architecture that combines simultaneously recurrent neural networks like vanilla RNN, LSTM or GRU with convolutional neural networks for graph-structured data. We have investigated here two architectures, one using a stack of CNN and RNN (Model 1), and one using convLSTM that considers convolutions instead of fully connected operations in the RNN definition (Model 2). We have then considered two applications: video prediction and natural language modeling. Model 2 has shown good performances in the case of video prediction, by improving the results of Shi et al. (2015). Model 1 has also provided promising performances in the case of language modeling, particularly in terms of learning speed. It has been shown that (i) isotropic filters, maybe surprisingly, can outperform classical 2D filters on images while requiring much less parameters, and (ii) that graphs coupled with graph CNN and RNN are a versatile way of introducing and exploiting side-information, e.g. the semantic of words, by structuring a data matrix.

Future work will investigate applications to data naturally structured as dynamic graph signals, for instance fMRI and sensor networks. The graph CNN model we have used is rotationally-invariant and such spatial property seems quite attractive in real situations where motion is beyond translation. We will also investigate how to benefit of the fast learning property of our system to speed up language modeling models. Eventually, it will be interesting to analyze the underlying dynamical property of generic RNN architectures in the case of graphs. Graph structures may introduce stability to RNN systems, and prevent them to express unstable dynamic behaviors.

## ACKNOWLEDGMENT

This research was supported in part by the European Union's H2020 Framework Programme (H2020-MSCA-ITN-2014) under grant No. 642685 MacSeNet, and Nvidia equipment grant.

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
