# Peer review of "Structured Sequence Modeling with Graph Convolutional Recurrent Networks"

_ICLR 2017 — rejected_

[Official Review · AnonReviewer2 · rating 4 · confidence 4 · 15 Dec 2016]
**Lacks technical novelty**

The authors address the problem of modeling temporally-changing signal on a graph, where the signal at one node changes as a function of the inputs and the hidden states of its neighborhood, the size of which is a hyperparameter. The approach follows closely that of Shi et al. 2015, but it is generalized to arbitrary graph structures rather than a fixed grid by using graph convolutions of Defferrard et al. 2016. This is not a strict generalization because the graph formulation treats all edges equally, while the conv kernels in Shi et al. have a built in directionality. The authors show results on a moving MNIST and on the Penn Tree Bank Language Modeling task.

The paper, model and experiments are decent but I have some concerns:

1. The proposed model is not exceptionally novel from a technical perspective. I usually don't mind if this is the case provided that the authors make up for the deficiency with thorough experimental evaluation, clear write up, and interesting insights into the pros/cons of the approach with respect to previous models. In this case I lean towards this not being the case.

2. The experiment results section is rather terse and light on interpretation. I'm not fully up to date on the latest of Penn Tree Bank language modeling results but I do know that it is a hotly contested and well-known dataset. I am surprised to see a comparison only to Zaremba et al 2014 where I would expect to see multiple other results.

3. The writing is not very clear and the authors don't make sufficiently strong attempt to compare the models or provide insight or comparisons into why the proposed model works better. In particular, unless I'm mistaken the word probabilities are a function of the neighborhood in the graph. What is the width of this graph? For example, suppose I sample a word in one part of the graph, doesn't this information have to propagate to the other parts of the graph along the edges? Also, it's not clear to me how the model can achieve reasonable results on moving MNIST when it cannot distinguish the direction of the moving edges. The authors state this but do not provide satisfying insight into how this can work. How does a pixel know that it should turn on in the next frame? I wish the authors thought about this more and presented it more clearly.

In summary, the paper has somewhat weak technical contribution, the experiments section is not very thorough, and the insights are sparse.

[Official Review · AnonReviewer1 · rating 4 · confidence 4 · 16 Dec 2016]
**Evaluation could be improved**

This paper investigates the  modeling of graph sequences . Authors propose Graph Convolutional Recurrent Networks (GRCN)  that extends convLSTM  (Shi et al. 2015) for data having an unregular graph structure at each timestep. They replace the 2D convolution with a graph convolutional operator from (Defferrad et al., 2016).
Authors propose two variations of the GRCN model. In Model 1,  the graph convolution is only applied on the input data. In Model 2, the graph convolution  is applied on both  input data and the previous hidden states. They evaluate their approaches on two different tasks, video generation using the movingMNIST dataset and world-level language modelling using Penntreebank.

On movingMNIST authors show that their GRCN 2 improves upon convLSTM. However, they evaluate only with one-layer convLSTM, while Shi et al. report better results with 3 layers (also not as good as  GRCN) . It would be nice to evaluate GCRCN in that setting as well.
While the authors show an improvement of GRCN relatively to convLSTM, GRCN on this task seems relatively weak compared to recent works such as the Video Pixel Networks (Kalchbrenner et al., 2016). It contradicts the claim that "Model 2 has shown good performance in the case of video prediction" in the conclusion.

For the Penntreebank experiments, author compares  their model 1 with FC-LSTM, with or without dropout. However, the results in (Zaremba et al., 2014) still seems different than the one reported here. In (Zaremba et al., 2014), they  reports a test perplexity of 78.4 for the large regularized LSTM in their table 1 which outperforms the score of the GRCN. Also, following works such as variational dropout or zoneout have since improve upon Zaremba results. Is there some differences in the experimental setting?  It would be nice to have results that are directly comparable to previous work.


Pros:
- Interesting model, 
Cons:
- Overall, the proposed contribution is relatively incremental compared to (Shi et al. 2015) and (Defferrad et al., 2016). 
- Weak results of GRCN relatively to previous works in the experiments, that do not convince of the GRCN advantages.

[Official Review · AnonReviewer3 · rating 4 · confidence 4 · 19 Dec 2016]

The paper proposes to combine graph convolution with RNNs to solve problems in which inputs are graphs. The two key ideas are: (i) a graph convolutional layer is used to extract features which are then fed in an RNN, and (ii) matrix multiplications are replaced by graph convolution operations. (i) is applied to language modelling, yielding lower perplexity on Penn Treebank (PTB) compared with LSTM. (ii) outperformed LSTM + CNN on the moving-MNIST.

Both two models/ideas are actually trivial and in line with the current trend of combining different architectures. For instance, the idea of replacing matrix multiplications by graph convolution is a small extension for Shi et al.

Regarding to the experiment on PTB (section 5.2), I'm skeptical about the way the experiment carried out. The reason is that, instead of using the given development set to tune the models, the authors blindly used an available configuration which is for a different model.

Pros: 
- good experimental results

Cons:
- ideas are quite trivial 
- the experiment on PTB was carried out improperly

[Final Decision · Program Chairs · 06 Feb 2017]
**ICLR committee final decision**

While graph structures are an interesting problem, as the reviewers observed, the paper extends previous work incrementally and the results are not very moving.
 
 pros
 - interesting problem space that has not been thoroughly explored
 cons
 - experimental evaluation was not convincing enough with the results.
 - the method itself is a small incremental improvement over prior papers.